# Effect of Protein Content on Heat Stability of Reconstituted Milk Protein Concentrate under Controlled Shearing

**DOI:** 10.3390/foods13020263

**Published:** 2024-01-14

**Authors:** Anushka Mediwaththe, Thom Huppertz, Jayani Chandrapala, Todor Vasiljevic

**Affiliations:** 1Advanced Food Systems Research Unit, Institute of Sustainable Industries & Liveable Cities and College of Sports, Health and Engineering, Victoria University, Werribee Campus, VIC 3030, Australia; anushka.mediwaththe@live.vu.edu.au (A.M.); or thom.huppertz@wur.nl (T.H.); 2FrieslandCampina, 3818 LE Amersfoort, The Netherlands; 3Food Quality and Design Group, Wageningen University & Research, 6808 WG Wageningen, The Netherlands; 4School of Science, RMIT University, Bundoora, VIC 3083, Australia; jayani.chandrapala@rmit.edu.au

**Keywords:** milk protein concentrate, protein concentration, heat stability, shear, aggregation

## Abstract

Milk protein concentrates (MPCs) possess significant potential for diverse applications in the food industry. However, their heat stability may be a limitation to achieving optimal functional performance. Shearing, an inherent process in food manufacturing, can also influence the functionality of proteins. The aim of this research was to examine the heat stability of reconstituted MPCs prepared at two protein concentrations (4% and 8% *w*/*w* protein) when subjected to varying levels of shearing (100, 1000, or 1500 s^−1^) during heating at 90 °C for 5 min or 121 °C for 2.6 min. While the impact of shear was relatively minor at 4% protein, it was more pronounced in 8% protein MPC suspensions, leading to a considerable decline in heat stability. An increase in protein concentration to 8% amplified protein interactions, intensified by shearing. This, in turn, resulted in comparatively higher aggregation at elevated temperatures and subsequently reduced the heat stability of the reconstituted MPCs.

## 1. Introduction

Milk protein concentrates (MPCs) are milk protein ingredients that contain whey proteins and caseins in their original proportions as found in milk [1,2,3]. They are highly valued for their high protein content, milky flavor, and ability to add opacity to beverages. MPCs have a diverse range of applications, including inclusion in recipes for production of high-protein dairy beverages, as well as cheese, yogurt, ice cream, infant formula, and health-related products [4,5,6,7,8,9]. In these applications, MPC powders are reconstituted and subjected to heat treatments, such as pasteurization, ultra-high-temperature processing, or retort sterilization, wherein their heat stability plays a crucial role [10,11,12]. Heat stability is a key factor in achieving optimal functional performance in food systems, influencing properties such as solubility, emulsification, foaming, viscosity, water binding, gelling, freeze–thaw ability, and acid stability, which are strongly interrelated [13,14,15,16]. It is essential to evaluate the heat stability of reconstituted MPC powders to ensure consistent processing outcomes and maintain product quality by preventing adverse effects caused by heat-induced destabilization.

Factors such as temperature and duration of heating, as well as exposure to mechanical forces during, e.g., stirring, pumping, and homogenization, along with compositional factors such as protein concentration and presence of various other additives such as sugars or salts, can contribute to heat-induced destabilization [17,18,19,20]. Under such conditions, proteins in MPCs may undergo substantial chemical and physical changes leading to aggregation, precipitation, sedimentation, or gelation, which can have detrimental effects on the quality of the final products.

Compositional factors play a significant role in heat-induced destabilization of milk and other milk protein-based systems [21]. It is widely recognized that the concentration of proteins affects the heat stability of milk protein systems. Concentrated milk typically exhibits lower heat stability compared to unconcentrated milk [22]. Similar observations have been made for MPCs, where greater protein concentration led to lower heat stability. This has been attributed to elevated levels of ionic calcium and increased viscosity [4,23,24]. Furthermore, the heat-induced formation of complexes between β-lactoglobulin (β-LG) and κ-casein (κ-CN) in the serum and colloidal phases of milk has been linked to heat stability issues [25,26].

Mechanical forces, such as shearing during the processing of products that contain MPCs, can also have a significant influence on the interactions between caseins and whey proteins. Shearing exposes protein structural elements to hydrodynamic shear stress, which can overcome stabilizing cohesive forces, such as intramolecular hydrogen bonds, resulting in protein unfolding [27,28,29,30]. Previous studies have indicated that shear has a substantial impact on the behavior of milk proteins across a wide range of temperatures [31].

Considering these findings, this study hypothesized that the heat stability of MPCs is profoundly influenced by the interplay of protein concentration, temperature, and shear. This hypothesis stemmed from the understanding that both heat and mechanical forces can cause substantial conformational changes in milk proteins, which in turn may affect the quality of the end product. Furthermore, there is a lack of research specifically examining the combined effects of temperature and shear on the heat stability of varying protein concentrations in MPCs. Therefore, the objective of this study was to investigate the influence of controlled shearing at different intensities (100, 1000, or 1500 s^−1^) and temperatures (121 °C for 2.6 min or 90 °C for 5 min) at two different protein concentrations (4% and 8%) of MPCs commonly used in the production of dairy beverages. By examining these parameters, the study aimed to provide insights into the impact of shear and temperature on the heat stability of MPCs with varying protein concentrations.

## 2. Materials and Methods

### 2.1. Materials

The MPC was acquired from Fonterra Co-operative (Palmerston North, New Zealand) and stored at −20 °C in airtight plastic containers. MPC powder contained 81.0% total protein, 1.6% fat, 5.5% carbohydrate, and 7.2% ash, according to the manufacturer’s declaration. All chemicals utilized in the analytical processes were sourced from Sigma-Aldrich Pty Ltd. (Castle Hill, NSW, Australia). Milli-Q water was used for all the experiments (Merck Millipore, Bayswater, VIC, Australia).

### 2.2. Preparation and Treatment of Samples

MPC suspensions with protein concentrations of 4% or 8% (*w*/*w*) were formulated by dissolving MPC powder in Milli-Q water. Each suspension was continuously stirred for 1 h at 50 °C for complete dispersion of the powder and continuously stirred at 4 °C overnight for complete hydration. Prior to commencing the experiments, the following day, the samples were allowed to equilibrate at a temperature of 25 °C for a duration of one hour [32]. Prepared MPC suspensions were heated at 121 °C for 2.6 min or 90 °C for 5 min at a constant shear rate of 0, 100, 1000, or 1500 s^−1^ in a pressure cell (CC25/PR-150) of a Physica MCR 301 series rheometer (Anton Paar GmbH, Ostfildern-Scharnhausen, Germany) with a constant pressure of 250 kPa using the method previously described [29]. Samples were heated at a rate of 5 °C min^−1^ to the required temperature and held there for the required time and cooled at a rate of 5 °C min^−1^.

### 2.3. Particle Size and Zeta Potential Measurements

Following each treatment, particle size and zeta potential measurements were performed using a Zetasizer Nano ZS (Malvern Instruments, Malvern, UK) as described previously [32].

### 2.4. Fourier Transform Infrared (FTIR) Analysis

The changes in protein secondary structural features were assessed by FTIR spectrometer (Frontier, PerkinElmer, Waltham, MA, USA) analysis as described previously [32].

### 2.5. Sodium Dodecyl Sulphate Polyacrylamide Gel Electrophoresis (SDS–PAGE)

After the treatments, a portion of all the treated and control samples was ultracentrifuged at 100,000× *g* for 1 h at 20 °C (Beckman Optima L-70 Ultracentrifuge, Indianapolis, IN, USA) to acquire the supernatant of the suspensions. Both non-reducing and reducing SDS–polyacrylamide gel electrophoresis (SDS–PAGE) were performed as described previously [32] for both bulk and supernatant MPC suspensions and stained with Coomassie Brilliant Blue (Sigma-Aldrich Pty Ltd., Castle Hill, NSW, Australia). The intensity of the treated reducing gel proteins in the supernatants was quantified as a percentage in relation to their corresponding protein content in the control bulk [32].

### 2.6. Determination of Protein Solubility and Heat Stability

The protein solubility at both concentrations was determined using the Kjeldahl method. After preparation of the suspensions as described in Section 2.2, these were centrifuged at 12,000× *g* for 20 min at 20 °C (model J2HS, Beckman, Fullerton, CA, USA) and supernatants were filtered through a 0.45 μm filter, following the procedure described previously [33]. A conversion factor of 6.38 for nitrogen was applied. Solubility was expressed as the protein content in the supernatant, relative to the total protein content in the original suspension, and expressed as a percentage [34,35].

The heat stability of each sample was evaluated by determining the protein content of the supernatant of samples before and after heat treatment. For this purpose, a portion of the sample was centrifuged at 12,000× *g* at 20 °C for 20 min (Model J2HS; Beckman, Fullerton, CA, USA) [34,35]. The protein content in the supernatants was determined using the Kjeldahl method, with a nitrogen conversion factor of 6.38. The heat stability of each suspension was then expressed based on these measurements as below:% Heat Stability = (Protein content of the supernatant of heated MPC suspension)/(Protein content of the supernatant of unheated MPC suspension) × 100%

### 2.7. Statistical Analysis

The experimental design was replicated at least three times with subsequent subsampling. Statistical analysis was carried out with the use of IBM SPSS statistics software version 28.0.1.0 (IBM Corp., Armonk, NY, USA) using a multivariate general linear model (GLM) protocol as a randomized, blocked, split plot in time with protein concentration as the main factor and temperature/time combination and shearing as subplots. The replications served as a block. The threshold for significance was established at *p* ≤ 0.05. The post hoc analysis for comparing multiple means was conducted using Tukey’s Studentized range test (HSD). All the results were reported as mean ± standard error. Principal component analysis (PCA) was conducted for FTIR analysis using PCA for spectroscopy app, Origin Pro 2018 (v.95E) in the broad Amide I region (1700–1600 cm^−1^) and in the region of 1200–900 cm^−1^ and score plots were obtained to group spectra for comparison purpose.

## 3. Results

### 3.1. Solubility and Heat Stability of Milk Protein Concentrations Subjected to Different Treatments

The solubility of MPC suspensions was comparable at both protein concentrations (Table 1). Heating both 4% and 8% protein MPC suspensions up to 121 °C resulted in a decline in heat stability, with both exhibiting similar levels at ~75% (Table 1).

The combined application of heat and shear exerted a noticeable impact on the heat stability of MPC suspensions. In the case of 4% protein MPC suspensions, heat stability decreased when subjected to low shear at 100 s^−1^ and at 90 °C down to ~80%. However, the heat stability increased to 83.1% when the shear rate was further raised to 1500 s^−1^. Heat stability appeared unaffected when shear was applied at 121 °C in 4% protein MPC suspensions (Table 1). The combination of heat and shear had a significant effect on the heat stability of 8% protein MPC suspensions, its reduction was apparent at both temperatures and clearly shear-dependent. The decline was more pronounced at 121 °C, where heat stability dropped to ~60% at a shear rate of 1500 s^−1^ in comparison to other conditions.

### 3.2. Average Particle Size and Zeta Potential Measurements of Milk Protein Concentrations Subjected to Different Treatments

The initial average particle size of 4% protein MPC suspensions was ~190 nm after hydration (Table 1). As the temperature increased to 90 °C, the average particle size decreased to ~174 nm and remained constant with further temperature elevation to 121 °C (Table 1). Similarly, the 8% protein MPC suspension initially had an average particle size of ~195 nm (Table 1). This size decreased to ~178 nm at 90 °C and showed no significant change with a further temperature increase to 121 °C (Table 1). The simultaneous application of heat and shear had no significant impact on the average particle size within both 4% and 8% protein concentrations at both 90 °C and 121 °C (Figure 1 and Table 1).

The zeta potential of the 4% protein MPC suspension was ~−21 mV, and heating did not induce significant changes (Table 1). However, for the 8% protein MPC suspension, heating had a notable effect, reducing the negative zeta potential from ~−23.7 mV at 20 °C to ~−22.7 mV at 121 °C (Table 1). Simultaneous application of heat and shear resulted in no substantial change in the zeta potential in 4% protein MPC suspensions at 90 °C. However, at 121 °C, the zeta potential became more negative at both 1000 s^−1^ and 1500 s^−1^ shear rates. At 1500 s^−1^ shear rate, the zeta potential increased to ~−26 mV from ~−22 mV at 121 °C with no shear. Conversely, in the 8% protein MPC suspensions, the zeta potential remained unchanged at both temperatures when shear was applied.

### 3.3. Secondary Structural Modifications of Milk Protein Concentrations Subjected to Different Treatments

In both 4% and 8% protein MPC suspensions, heating at 90 °C resulted in primarily slight variations in β-sheets and α-helices (Figure 2 and Table 2). The structural changes were more pronounced at 8% protein, with a reduction down to ~68% in α-helices and an increase up to ~67% in β-sheets at 121 °C, indicative of extensive aggregation. In comparison, at 4% protein, there was a reduction down to ~41% in α-helices and an increase up to ~62% in β-sheets at 121 °C. This was also accompanied by a substantial increase in β-turns (~1667–1684 cm^−1^) and a decrease in random structures (~1643–1646 cm^−1^) at 8% concentration (Figure 2 and Table 2).

The application of shear-induced protein secondary structural changes, with the extent of these changes dependent on the applied conditions as well as protein content within suspensions (Figure 2 and Table 2). The combined application of heat and shear resulted in noticeable changes in the content of β-sheets in 4% protein suspensions. At 90 °C, intense peaks at ~1688 cm^−1^ were observed at a shear rate of 100 s^−1^, demonstrating intermolecular and anti-parallel β-sheet-driven aggregation. These peaks were comparatively less pronounced at shear rates of 1000 s^−1^ and 1500 s^−1^, suggesting a dominance of fragmentation of aggregates [36] (Figure 2 and Table 2). A further increase in temperature to 121 °C revealed the prevalence of shear-induced fragmentation. An increase in shear up to 1500 s^−1^ resulted in a ~50% increase in random structures, suggesting the dominant fragmentation of aggregates at 121 °C (Table 2).

On the other hand, the combined application of heat and shear to 8% protein MPC suspensions resulted in considerable β-sheet-driven aggregation at 90 °C, peaking at 1500 s^−1^, resulting in a 32% increase in β-sheet content [36]. Furthermore, reduced peaks at ~1675 cm^−1^ and ~1624 cm^−1^ denote a reduced presence of β-turns and decreased intramolecular crosslinking consequently [37] (Figure 2 and Table 2). Further heating up to 121 °C in combination with shear also revealed the prevalence of shear-induced aggregation in 8% protein suspensions. An intense peak at ~1655 cm^−1^ at 1500 s^−1^ at 121 °C denotes possible shear-induced reformation of α-helical structures (Figure 2). Total β-sheets increased by ~13% at 1500 s^−1^, suggesting further aggregation of proteins at 121 °C (Table 2).

These variations in the second derivative spectra of the Amide I region were further analyzed using principal component analysis (PCA). The PCA separated samples based on the shear rates at different concentrations and temperatures. However, no distinguishable separation of spectra was observed, suggesting subtle changes in secondary structural modifications that did not significantly contribute to the overall variance in the dataset (Appendix A).

To explore the effects of treatments on the behavior of lactose and minerals, particularly phosphate under the given conditions, the FTIR spectral region of 1200–900 cm^−1^ was also subjected to analysis. In 4% protein suspensions, distinguishable peaks were observed at ~970 cm^−1^ and at ~990 cm^−1^ at both 1000 s^−1^ and 1500 s^−1^ and at 90 °C (Figure 3). When temperature was further increased to 121 °C, two prominent peaks at ~970 cm^−1^ and ~987 cm^−1^ were observed at 1500 s^−1^.

Similar to 4% protein suspensions, prominent peaks were detected at both ~970 cm^−1^ and 990 cm^−1^ in 8% protein suspensions sheared at 1500 s^−1^ and 90 °C within the region of 1200–900 cm^−1^. At 121 °C, a distinguishable peak ~990 cm^−1^ was observed at 1500 s^−1^ (Figure 3). These observations were further supported by PCA analysis performed in the region of 1200–900 cm^−1^, which explained over 85% of the variance and captured most of the spectral changes.

### 3.4. Partitioning of Proteins in Milk Protein Concentrate Subjected to Different Treatments

Protein interactions were further analyzed using non-reducing and reducing SDS–PAGE. In both 4% and 8% protein, the presence of aggregates was observed in all treatments at both 90 °C and 121 °C. These aggregates, which were observed on top of non-reducing stacking gels, completely disappeared under reducing conditions, indicating that they were exclusively formed as a result of thiol/disulfide interactions (Figure 4 and Figure 5). Heating both 4% and 8% protein MPC suspensions resulted in a gradual decrease in β-LG and α-LA concentrations, with no notable difference between the two concentrations (Table 3). Conversely, all caseins (α_s_-CN, β-CN, and κ-CN) in the supernatant increased, and there was no significant difference observed between the two concentrations (Table 3).

The application of heat and shear resulted in an increased concentration of β-LG, α-LA, and κ-CN in the supernatants of both 4% and 8% protein MPC suspensions heated at both 90 °C and 121 °C, compared to the suspensions subjected solely to heat treatment. Notably, this increase was more pronounced in the 4% protein suspensions than in the 8% protein MPC suspensions (Table 3). Particularly, κ-CN exhibited a higher dissociation in 4% protein suspensions compared to 8% protein suspensions at the highest shear rate of 1500 s^−1^ at 121 °C.

In 4% protein suspensions, at 121 °C, increasing the shear from 0 s^−1^ to 1500 s^−1^ resulted in a ~23% increase in β-LG, a ~22% increase in α-LA, and a ~23% increase in κ-CN levels in the supernatant. On the other hand, in 8% protein suspensions at 121 °C, raising the shear from 0 s^−1^ to 1500 s^−1^ led to a ~9% increase in β-LG, a ~3% increase in α-LA, and a ~6% increase in κ-CN levels in the supernatant (Table 3).

## 4. Discussion

In both 4% and 8% protein MPC suspensions, micellar casein dissociation and the formation of smaller whey protein aggregates with an increase in temperature were evident, as observed by SDS–PAGE data showing elevated levels of soluble caseins and a decrease in whey proteins in the soluble phase (Table 3). However, the level of aggregation during heating did not appear to be influenced by the protein content at either temperature. The micellar casein dissociation and formation of smaller whey protein aggregates may have led to a reduction in average particle size at both concentrations during heating [37] (Table 1). However, this reduction in particle size with heating was not significantly affected by the heating temperature or the protein content within the suspension. Additionally, while the heat stability gradually decreased as the temperature increased at both 4% and 8% protein concentration, it was not influenced by the protein content, especially at 121 °C, where comparable levels were reached in both suspensions (Table 1). These observations were further supported by FTIR analysis, where no noticeable difference in the extent of heat-induced changes was observed in both concentrations that could be attributed to their protein content, as evidenced by PCA analysis. These findings align with previous studies by [38], where the rate constants for the thermal denaturation of β-LG were known to be independent of the initial protein concentration in skim milk systems. Notably, it was established that, at higher temperatures (>~90 °C), the rate constants remained independent of the whey protein concentration [38,39,40]. This implies that the heat stability of proteins was not markedly influenced by the initial protein concentration under the given conditions. The fact that the denaturation rate constant remained unaffected by the initial protein concentration may contribute to the observed uniformity in heat stability. However, it is important to note that the rate of denaturation, while a factor, may not be the primary influence on heat stability. Other factors or interactions could also be influential in determining the overall stability of the system subjected to heat treatment [23].

The combined application of heat and shear resulted in increased levels of β-LG, α-LA, and κ-CN in the soluble phase, indicating significant shear-induced fragmentation of aggregates in both suspensions (Table 3). Fluid dynamics play a crucial role in affecting aggregation kinetics and can introduce disruptive stresses that lead to the breakdown of agglomerates [41,42,43,44,45]. This shear-induced fragmentation occurs through controlled fragmentation of complexes due to pressure fluctuations in fluid flow, particle fragmentation from larger complexes, and abrasion of primary complexes from newly created particle surfaces [46,47]. In the case of 4% protein suspension, higher levels of β-LG, α-LA, and κ-CN were observed in the soluble phase, indicating a pronounced impact of shear on cohesive forces, including hydrophobic, van der Waals, or electrostatic interactions between particles (Table 3). The FTIR data in 4% protein MPC suspensions provided further evidence of shear-induced fragmentation, as shown by reduced peak intensities of intermolecular β-sheets and the presence of unordered structures, particularly at high shear rates of 1000 s^−1^ and 1500 s^−1^ (Figure 2 and Table 2). Furthermore, the dissociation of caseins was prominent in 4% protein MPC suspensions when subjected to shear, as supported by FTIR results, which showed two concurrent peaks at ~990 cm^−1^ and ~980 cm^−1^ at both temperatures (Figure 3). The appearance of the latter band may indicate the release of the MCP nanoclusters from the casein micelle, resulting in an increased negative charge of the casein molecules, particularly observed at 121 °C, where there was a substantial increase in negative zeta potential at both 1000 s^−1^ and 1500 s^−1^ shear rates compared to control (Table 1). The presence of the former band suggests the dissociation of the MCP into Ca^2+^ and HPO_4_^2−^ upon release from the serine phosphate residues. The intensities of these two bands correlate with the micelle dissociation process. In addition, this was supported by SDS–PAGE analysis, revealing a prominent increase in κ-CN content in the soluble phase at the highest shear rate of 1500 s^−1^ at 121 °C in 4% protein suspensions, which could be attributed to shear-induced casein dissociation (Table 3). Shear flow exerts fluid drag, disrupting and destabilizing casein micelles, making them more susceptible to dissociation, as observed in previous studies [31,48]. This shear-induced micellar dissociation and fragmentation of aggregates were further supported by particle size data, demonstrating a reduction in particle size, especially at 121 °C, suggesting a more pronounced effect (Table 1). However, in 4% protein suspensions, heat stability decreased at 100 s^−1^ and 90 °C. A further increase in shear up to 1500 s^−1^ increased the heat stability. Nevertheless, heat stability was unaffected by increased shear at 121 °C. As observed, applied shear can disrupt protein components and modify their structure, depending on its magnitude, leading to alterations in the distribution of surface-active proteins within the concentrate [49,50] and contributing to variations in the heat stability of MPC suspensions.

When compared to 4% protein suspensions, shear-induced fragmentation was observed to be less pronounced in 8% protein MPC suspensions. This could be due to either resistance to applied shear forces dominating as cohesive stabilization or shear-induced aggregation becoming dominant up to a certain extent. The process of aggregation, break-up, and restructuring in a dispersion is intricately influenced by the balance between hydrodynamic forces and cohesive forces [51,52,53,54]. At high shear rates, hydrodynamic drag forces disrupt cohesive forces, such as intramolecular hydrogen bonds, leading to protein unfolding. Consequently, exposed reactive sites facilitate the formation of complexes between whey proteins and micellar caseins, resulting in further aggregation at elevated temperatures [55,56,57]. As the milk protein concentration increases, the likelihood of protein–protein interactions rises, leading to protein aggregation and the formation of larger complexes due to the influence of shear. The prominent aggregation with an increase in shear in 8% protein suspensions is evident from the FTIR data, which indicates an increase in intermolecular β-sheets as a result of β-sheet-driven aggregation, along with a decrease in random structures, particularly at 121 °C, compared to heated controls (Table 2) [58]. In addition, comparatively less release of β-LG, α-LA, and κ-CN into the soluble phase (Table 2), as well as no observable effect on particle size with an increase in shear compared to 4% protein suspensions (Table 1), specifically explains a certain level of aggregation in 8% protein MPC suspensions compared to 4% protein suspensions. Furthermore, the stabilizing effect against heat-induced changes may become less effective due to the effect of hydrodynamic shear, which improves particle collisions. The growth rate and size of protein aggregates are influenced by the balance between growth and controlled shear-induced breakage. Shear-induced growth (orthokinetic aggregation) in shear flow directly correlates with shear rate, particle quantity, and capture efficiency, with the latter linked to medium viscosity [59,60,61,62]. Owing to its higher total solid content and increased particle concentration, 8% protein MPC suspension exhibits elevated viscosity compared to the 4% protein MPC suspension. This higher viscosity further enhances the system’s resistance to shear forces, subsequently modulating particle interactions. Consequently, the 8% protein MPC suspension demonstrates pronounced aggregation in contrast to the 4% protein MPC suspension, which experiences fragmentation. The larger complexes that are formed can disrupt the original structure, rendering the system more susceptible to coagulation, gelation, and other forms of destabilization when exposed to heat, ultimately resulting in reduced heat stability. Reduced heat stability was observed at both 90 °C and 121 °C, along with the increase in shear, resulting in the lowest levels at 1500 s^−1^ and at 121 °C (Table 1). Collectively, these multifaceted factors collaborate to govern the ultimate size, structural configuration, and stability of particles within the given system.

## 5. Conclusions

Shearing has the ability to modify the structure and physicochemical properties of milk proteins, which, in turn, can affect the mechanisms of protein interactions and aggregation during heating and impact heat stability. When comparing 8% protein MPC suspensions to 4% protein suspensions, it was observed that higher shear levels result in lower heat stability in 8% protein suspensions. The increase in protein concentration up to 8% enhances protein interactions, making proteins more prone to aggregate formation during high-temperature heating. At high shear rates, cohesive stabilizing forces, such as intramolecular hydrogen bonds that maintain the helical structure, are disrupted by hydrodynamic drag forces, leading to molecular unfolding. As a result, previously hidden reactive sites are exposed, promoting the formation of complexes between whey proteins and casein micelles. This, in turn, leads to further aggregation at high temperatures and ultimately reduces heat stability. Therefore, considering the given parameters, low protein content (4%) and low temperatures would be most suitable for providing optimal heat stability under high shear conditions in reconstituted milk products. In addition, it is known that commercial MPCs vary greatly in terms of their gross composition including protein and mineral content, which can also govern the course of denaturation and/or aggregation of proteins upon heating and shearing. For this reason, a wider range of commercial MPCs should be assessed in order to generalize the observations and conclusions of this study.

## Figures and Tables

**Figure 1 foods-13-00263-f001:**
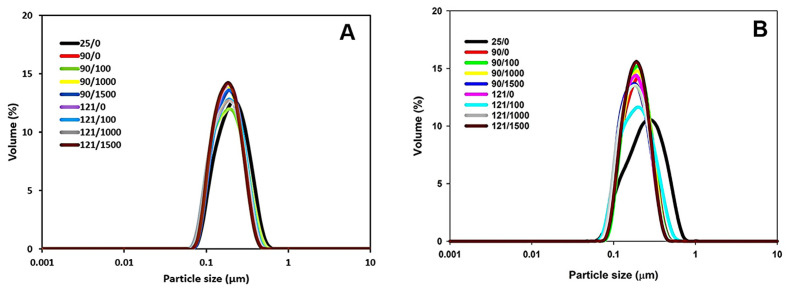
Particle size distribution of 4% protein MPC suspension subjected to heating at 90 or 121 °C and shear rate of 100, 1000, or 1500 s^−1^ (**A**) or 8% protein MPC suspension subjected to heating at 90 or 121 °C and shear rate of 100, 1000, or 1500 s^−1^ (**B**). The true controls were assessed prior to heating without shear (0 s^−1^).

**Figure 2 foods-13-00263-f002:**
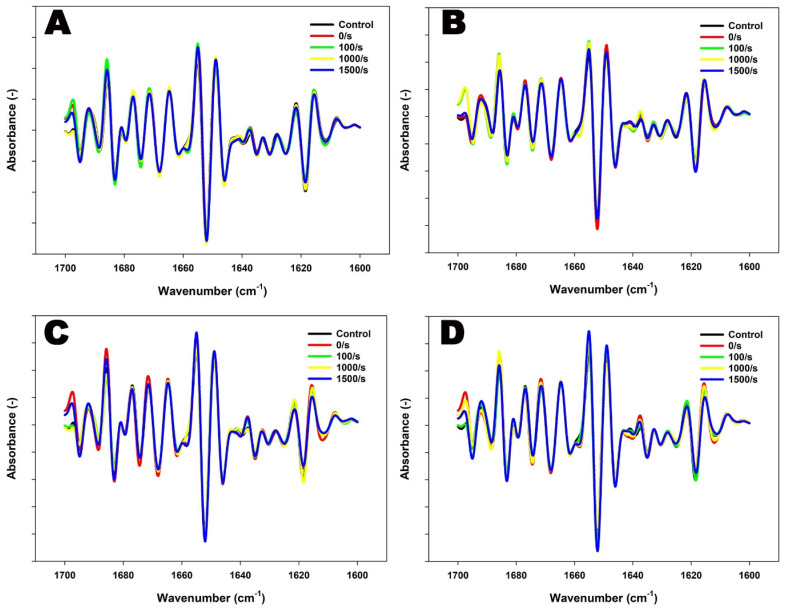
FTIR spectra (second derivative) for the Amide I region of 4% protein MPC suspensions sheared at 100, 1000, or 1500 s^−1^ during heating at 90 °C for 5 min (**A**) or 121 °C for 2.6 min (**B**), and 8% protein MPC suspensions sheared at 100, 1000, or 1500 s^−1^ during heating at 90 °C for 5 min (**C**) or 121 °C for 2.6 min (**D**). The true controls were assessed prior to heating without shear (0 s^−1^).

**Figure 3 foods-13-00263-f003:**
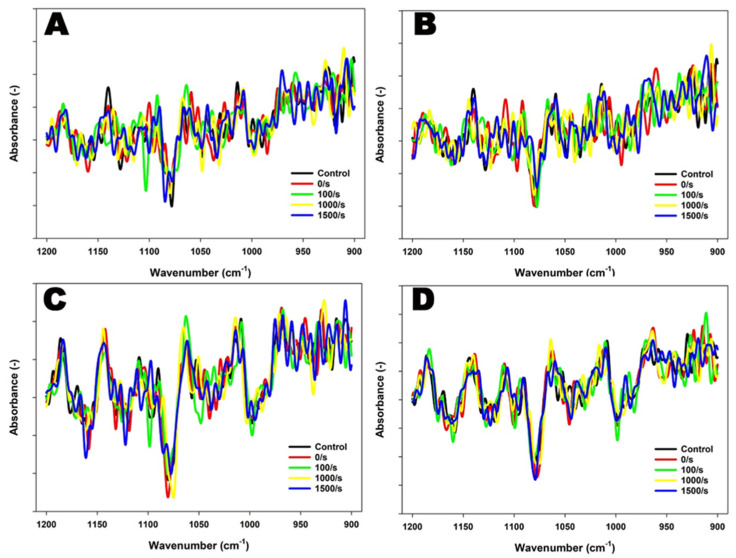
FTIR spectra of 4% protein MPC suspensions sheared at 100, 1000, or 1500 s^−1^ during heating at 90 °C for 5 min (**A**) or 121 °C for 2.6 min (**B**), and 8% protein MPC suspensions sheared at 100, 1000, or 1500 s^−1^ during heating at 90 °C for 5 min (**C**) or 121 °C for 2.6 min (**D**), obtained in the region between 1200 and 900 cm^−1^. The true controls were assessed prior to heating without shear (0 s^−1^).

**Figure 4 foods-13-00263-f004:**
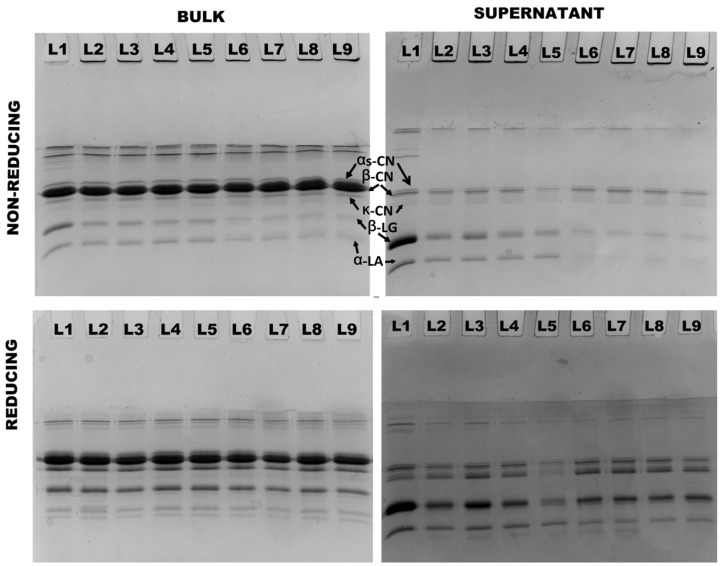
Non-reducing and reducing SDS–PAGE analysis of bulk and supernatant of 4% protein MPC suspensions (Lane sequence—(25-0)/90-0/90-100/90-1000/90-1500/121-0/-121-100/121-1000/121-1500 from left to right).

**Figure 5 foods-13-00263-f005:**
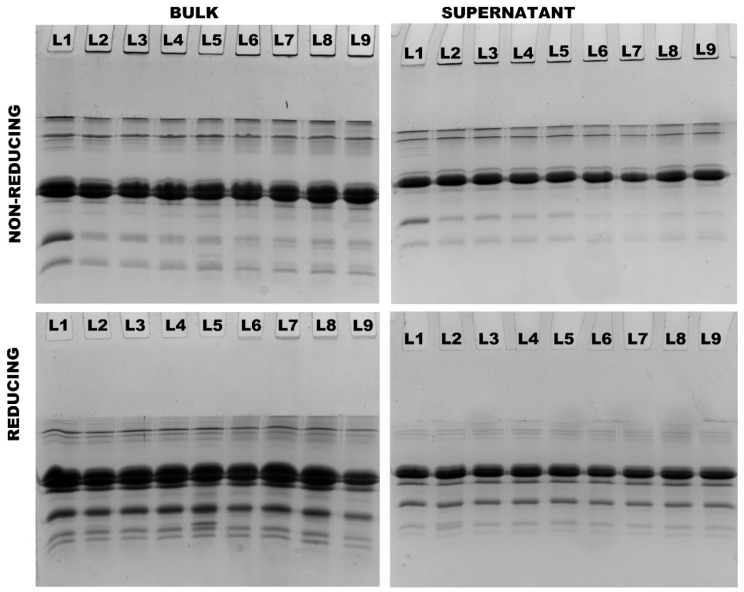
Non-reducing and reducing SDS–PAGE analysis of bulk and supernatant of 8% protein MPC suspensions (Lane sequence—(25-0)/90-0/90-100/90-1000/90-1500/121-0/-121-100/121-1000/121-1500 from left to right).

**Table 1 foods-13-00263-t001:** Particle size, zeta potential, solubility, and heat stability of 4% and 8% protein MPC suspensions subjected to shearing at 90 °C for 5 min or 121 °C for 2.6 min.

Protein (%)	Shear Rate (s^−1^)	Average Particle Size (nm)			Zeta Potential (mV)			Solubility %	Heat Stability %	
		25 °C	90 °C	121 °C	25 °C	90 °C	121 °C		90 °C	121 °C
4	0	190 ± 2 ^Ba^	174 ± 1 ^Bb^	177 ± 1 ^Ab^	−21.1 ± 0.1 ^Ba^	−22.7 ± 0.6 ^Aa^	−21.7 ± 0.5 ^Ca^	98.7 ± 0.1 ^A^	85.5 ± 0.1 ^Aa^	75.0 ± 0.9 ^ABb^
	100		177 ± 1 ^ABa^	176 ± 2 ^Aa^		−22.0 ± 0.4 ^Aa^	−22.7 ± 0.5 ^Ca^		80.1 ± 0.6 ^BCa^	74.6 ± 1.0 ^ABb^
	1000		177 ± 2 ^ABa^	175 ± 2 ^Aa^		−22.6 ± 0.5 ^Ab^	−24.2 ± 0.1 ^Ba^		81.6 ± 0.3 ^BCa^	78.2 ± 0.9 ^Aa^
	1500		178 ± 2 ^ABa^	175 ± 1 ^Aa^		−21.3 ± 0.8 ^Ab^	−25.9 ± 0.2 ^Aa^		83.1 ± 0.1 ^ABa^	80.1 ± 3.1 ^Aa^
8	0	195 ± 1 ^Aa^	178 ± 2 ^ABb^	176 ± 2 ^Ab^	−23.7 ± 0.2 ^Ab^	−22.6 ± 0.5 ^Aa^	−22.7 ± 0.5 ^Ca^	95.4 ± 0.3 ^B^	79.0 ± 1.1 ^Ca^	75.0 ± 0.8 ^ABb^
	100		178 ± 2 ^ABa^	175 ± 1 ^Aa^		−21.7 ± 0.5 ^Aa^	−22.1 ± 0.4 ^Ca^		75.3 ± 0.3 ^Da^	70.2 ± 0.6 ^BCb^
	1000		177 ± 1 ^ABa^	175 ± 1 ^Aa^		−21.6 ± 0.6 ^Aa^	−22.8 ± 0.6 ^Ca^		70.3 ± 0.5 ^Ea^	65.6 ± 0.4 ^CDb^
	1500		179 ± 1 ^Aa^	176 ± 2 ^Aa^		−21.5 ± 0.6 ^Aa^	−22.5 ± 0.5 ^Ca^		65.4 ± 0.1 ^Fa^	60.0 ± 0.4 ^Db^

The values are presented as means of subsampling from three independent observations, ±standard error. Values denoted with different uppercase letters within columns and different lowercase letters within rows for the same parameter indicate significant differences (*p* < 0.05).

**Table 2 foods-13-00263-t002:** Total percentage areas of different secondary structures in the Amide I region of proteins in 4% and 8% protein MPC suspensions subjected to heat and shear treatments.

Protein (%)	Temp. (°C)	Shear Rate (s^−1^)	α-Helix (1646–1664 cm^−1^)	Total β-Sheet (1615–1637/1682–1700 cm^−1^)	Total β-Turns (1665–1681 cm^−1^)	Random (1638–1645 cm^−1^)
4	25	0	5.4 ± 0.2 ^e^	43.1 ± 0.1 ^n^	8.9 ± 0.1 ^j^	42.6 ± 0.3 ^b^
	90	0	4.1 ± 0.1 ^h^	47.2 ± 0.3 ^l^	7.4 ± 0.4 ^k^	41.3 ± 0.1 ^c^
		100	2.0 ± 0.3 ^l^	52.7 ± 0.4 ^i^	6.4 ± 0.1 ^m^	38.9 ± 0.2 ^d^
		1000	3.8 ± 0.5 ^i^	46.8 ± 0.2 ^m^	12.0 ± 0.1 ^e^	37.4 ± 0.1 ^e^
		1500	7.7 ± 0.1 ^b^	48.8 ± 0.2 ^k^	12.2 ± 0.1 ^e^	31.3 ± 0.4 ^f^
	121	0	3.2 ± 0.1 ^j^	69.9 ± 0.3 ^b^	15.8 ± 0.4 ^b^	11.1 ± 0.3 ^p^
		100	5.4 ± 0.4 ^e^	67.4 ± 0.1 ^d^	13.3 ± 0.2 ^d^	13.9 ± 0.5 ^n^
		1000	4.8 ± 0.1 ^f^	70.9 ± 0.4 ^a^	11.2 ± 0.7 ^f^	13.1 ± 0.1 ^o^
		1500	4.7 ± 0.2 ^fg^	68.5 ± 0.1 ^c^	10.2 ± 0.1 ^h^	16.6 ± 0.7 ^m^
8	25	0	8.0 ± 0.4 ^b^	35.8 ± 0.1 ^o^	6.8 ± 0.5 ^l^	49.4 ± 0.1 ^a^
	90	0	6.2 ± 0.1 ^d^	50.4 ± 0.2 ^j^	15.2 ± 0.1 ^c^	28.2 ± 0.4 ^g^
		100	10.0 ± 0.7 ^a^	57.8 ± 0.2 ^h^	10.4 ± 0.3 ^h^	21.8 ± 0.3 ^j^
		1000	7.2 ± 0.1 ^c^	61.8 ± 0.2 ^f^	10.0 ± 0.3 ^i^	21.0 ± 0.1 ^k^
		1500	7.1 ± 0.8 ^c^	66.4 ± 0.5 ^e^	7.1 ± 0.1 ^kl^	19.4 ± 0.3 ^l^
	121	0	2.6 ± 0.1 ^k^	59.7 ± 0.2 ^g^	17.9 ± 0.2 ^a^	19.8 ± 0.2 ^l^
		100	2.8 ± 0.3 ^jk^	57.6 ± 0.2 ^h^	16.0 ± 0.1 ^b^	23.6 ± 0.1 ^i^
		1000	4.4 ± 0.2 ^fgh^	61.2 ± 0.1 ^f^	10.8 ± 0.3 ^g^	23.6 ± 0.3 ^i^
		1500	4.3 ± 0.1 ^gh^	67.2 ± 0.2 ^d^	3.9 ± 0.1 ^n^	24.6 ± 0.1 ^h^

The data are represented as mean values ± standard error, derived from a minimum of three independent replications. Significant differences within a column are denoted by different lower-case superscript letters at *p* < 0.05.

**Table 3 foods-13-00263-t003:** Intensity of individual caseins, β-LG, and α-LA in supernatants of 4% and 8% protein MPC suspensions as a % of their intensity in the respective control bulk suspensions subjected to different treatments resolved under reducing electrophoretic conditions and quantified using a ChemiDoc imager.

Protein (%)	Temp. (°C)	Shear Rate (s^−1^)	α_s_-CN	β-CN	κ-CN	β-LG	α-LA
4	25	0	7.5 ± 0.1 ^h^	9.8 ± 0.4 ^f^	28.7 ± 1.2 ^i^	99.3 ± 0.6 ^a^	99.7 ± 0.1 ^a^
	90	0	8.1 ± 0.3 ^f^	10.3 ± 0.1 ^e^	31.3 ± 0.9 ^h^	55.1 ± 0.8 ^f^	63.8 ± 0.9 ^f^
		100	8.8 ± 0.2 ^d^	9.7 ± 0.7 ^f^	38.2 ± 0.4 ^e^	56.2 ± 0.3 ^f^	64.2 ± 0.7 ^f^
		1000	8.3 ± 0.1 ^e^	9.2 ± 0.6 ^g^	45.2 ± 0.7 ^d^	67.8 ± 0.4 ^d^	71.3 ± 0.4 ^d^
		1500	7.8 ± 0.4 ^g^	8.9 ± 0.2 ^h^	48.5 ± 0.5 ^c^	75.0 ± 0.2 ^b^	78.3 ± 0.5 ^c^
	121	0	10.4 ± 0.2 ^b^	12.5 ± 0.1 ^a^	35.3 ± 0.2 ^f^	50.4 ± 0.9 ^g^	59.3 ± 0.2 ^h^
		100	11.3 ± 0.7 ^a^	11.8 ± 0.9 ^b^	37.8 ± 0.4 ^e^	51.5 ± 1.1 ^g^	61.5 ± 0.5 ^g^
		1000	10.5 ± 0.9 ^b^	10.9 ± 1.1 ^c^	51.3 ± 0.5 ^b^	70.3 ± 0.7 ^b^	79.8 ± 0.2 ^c^
		1500	9.8 ± 0.2 ^c^	10.6 ± 0.5 ^d^	58.2 ± 0.1 ^a^	73.8 ± 0.5 ^b^	81.3 ± 0.4 ^b^
8	25	0	6.8 ± 0.2 ^k^	8.1 ± 0.5 ^i^	20.3 ± 0.5 ^k^	98.7 ± 0.2 ^a^	99.1 ± 0.6 ^a^
	90	0	7.2 ± 0.1 ^i^	9.3 ± 0.3 ^g^	23.4 ± 0.1 ^j^	58.3 ± 0.9 ^e^	65.1 ± 0.5 ^f^
		100	7.0 ± 0.5 ^j^	7.3 ± 0.2 ^j^	29.7 ± 0.9 ^i^	57.5 ± 0.7 ^ef^	63.8 ± 0.3 ^f^
		1000	5.3 ± 0.3 ^m^	6.5 ± 0.7 ^l^	32.5 ± 0.7 ^g^	59.1 ± 0.4 ^e^	68.1 ± 0.4 ^e^
		1500	4.9 ± 0.4 ^n^	5.9 ± 0.2 ^m^	33.8 ± 0.8 ^g^	68.5 ± 0.5 ^cd^	73.2 ± 0.9 ^d^
	121	0	10.5 ± 0.8 ^b^	10.3 ± 0.7 ^e^	30.8 ± 0.5 ^hi^	47.3 ± 0.3 ^h^	58.5 ± 0.4 ^h^
		100	9.8 ± 0.5 ^c^	10.2 ± 0.2 ^e^	31.3 ± 0.4 ^h^	45.8 ± 0.2 ^h^	56.3 ± 0.1 ^i^
		1000	7.5 ± 0.3 ^h^	6.8 ± 0.7 ^k^	35.7 ± 0.2 ^f^	52.3 ± 1.3 ^g^	58.5 ± 1.3 ^h^
		1500	6.1 ± 0.2 ^l^	5.3 ± 0.3 ^n^	37.2 ± 0.5 ^e^	55.8 ± 0.9 ^f^	61.8 ± 0.4 ^g^

Values represent the means derived from a minimum of three independent replications, ±standard error. Significant differences within a column are denoted by different lower-case superscript letters at *p* < 0.05.

## Data Availability

Data is contained within the article or Appendix A.

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
