# Peer review of "Effect of Protein Content on Heat Stability of Reconstituted Milk Protein Concentrate under Controlled Shearing"

_foods, 2024, doi:10.3390/foods13020263_

Round 1
Reviewer 1 Report
Comments and Suggestions for Authors
The authors examined the effect of 4% and 8% w/w protein concentration on the heat stability of reconstituted milk protein concentrates when subjected to varying levels of shearing (100, 1000, and 1500s -1) during heating at 90℃ for 5 min or 121℃ for 2.6 min. This study belongs to a basic research and is very interesting. The results were discussed well and can inspire the colleague. Suggestions,
1) In all the tables, the standard errors had better be added.
2) In table 3, the caseins and beta-LG, alpha-LA should be noted.
3) Line 220-225, the PCA figures should be given.
4) Figs. S1 to S5 had better insert to the whole text. From figs S1 to S5, the typical treatments should be shown and thus the colleague can be easy to read. In figures S4 and S5, the protein strips in SDS-PAGE with silver staining were not neat and beautiful due to the lower salt concentration in samples.
5) The better combination of protein concentration, temperature, and shear rate should be suggested to reconstituted milk protein products.
Author Response
|
Editor’s/Reviewer’s comments |
Authors’ Response – Details of Amendments to Manuscript |
Line no. / Font Colour |
|
Reviewer #1 |
|
|
|
1. The authors examined the effect of 4% and 8% w/w protein concentration on the heat stability of reconstituted milk protein concentrates when subjected to varying levels of shearing (100, 1000, and 1500s -1) during heating at 90℃ for 5 min or 121℃ for 2.6 min. This study belongs to basic research and is very interesting. The results were discussed well and can inspire the colleague. |
Thank you for your valuable comments. We truly appreciate your recognition of our study as highly interesting and inspiring to colleagues. |
|
|
2. In all the tables, the standard errors had better be added. |
Thank you for your valuable comment. Standard errors have been added to all the tables as suggested. |
|
|
3. In table 3, the caseins and beta-LG, alpha-LA should be noted. |
Title of the table 3 was revised as suggested. |
L286-289 |
|
4. Line 220-225, the PCA figures should be given. |
The PCA figures, referenced in lines 220-225 which are related to the Amide I region, have been added as Supplementary Figure 1. |
|
|
5. Figs. S1 to S5 had better insert to the whole text. From figs S1 to S5, the typical treatments should be shown and thus the colleague can be easy to read. In figures S4 and S5, the protein strips in SDS-PAGE with silver staining were not neat and beautiful due to the lower salt concentration in samples. |
Figs S1 to S5 have been inserted to the whole text as suggested.
While we acknowledge that the gels could have been clearer the staining was achieved using the Coomasse blue stain – we have added this |
|
|
6. The better combination of protein concentration, temperature, and shear rate should be suggested to reconstituted milk protein products. |
Thanks, the conclusion section has now been revised to suggest a better combination of protein concentration, temperature, and shear for reconstituted milk products. |
L417-419 |
Reviewer 2 Report
Comments and Suggestions for Authors
Topic of manuscript is interesting and relevant for the field and is within the section of “Dairy” in Foods journal. I have listed below the major weaknesses of this manuscript:
1. The experiment repetition and multiple comparison methods should be explained, and it is recommended to present the data in the form of mean ± standard deviation.
2. Line 154, "without significant (P > 0.05) change observed upon further increase of shear," that are inconsistent with the significance results in Table 1.
3. The SDS-PAGE should include a standard protein marker.
4. It is suggested that some of the typical figures from the Supplementary file be included in the manuscript.
5. The references in the introduction and discussion sections need to be updated. The manuscript only includes 6 references from the past 5 years, and the reference format needs to be standardized.
6. The graphical abstract is very roughly drawn and needs to be re-layout and revised.
Author Response
For responses please see the attached file

Reviewer 3 Report
Comments and Suggestions for Authors
Although the topic is relevant and the approach is sound mainly, some areas need clarification, expansion, or re-examination.
1. Introduction:
Hypothesis/es missing.
Explain in the background why the purpose of the study is considered essential and its relevance for a wider audience.
2. Methodological questions:
- How was the sample size determined? Could the result be generalisable? Why readers should read results that cannot be generalised. At the very least, this should be included in the study's limitations.
3. Discussion and conclusion:
Please improve the discussion of implacability and insert limitations of the study.
Author Response
|
Editor’s/Reviewer’s comments |
Authors’ Response – Details of Amendments to Manuscript |
Line no. / Font Colour |
|
Reviewer #3 |
|
|
|
1. Although the topic is relevant and the approach is sound mainly, some areas need clarification, expansion, or re-examination. |
Thank you. We sincerely value your feedback, which helps us to improve our manuscript. |
|
|
2. Introduction: Hypothesis/es missing. Explain in the background why the purpose of the study is considered essential and its relevance for a wider audience. |
Thanks, the hypothesis has now been included in the introduction section to enhance clarity. |
L69-73 |
|
3. Methodological questions: How was the sample size determined? Could the result be generalisable? Why readers should read results that cannot be generalised. At the very least, this should be included in the study's limitations. |
We have followed the standard research protocol including the replication and consequent subsampling as we indicated in the manuscript. We also assessed the data using a multivariate GLM model. While a power analysis can be used to determine the sample size, in this particular case it was not applied due to inability to ensure compositional consistency. Due to a large number of commercially available MPCs on the market we were not able to assess them all as the compositional differences may vary greatly and that would influence the outcomes. For this reason, the study is limited to the sample assessed and some generalisation could be drawn. We have included a statement in the conclusion to address this limitation. |
L421-426 |
|
4. Discussion and conclusion: Please improve the discussion of implacability and insert limitations of the study. |
See the comment above, we have addressed this by inclusion of a statement on the implications and limitations of the study. |
L421-426 |
|
Editor’s comments |
|
|
|
1. We find that the repetition rate in this manuscript is too high, please revise your manuscript according to the *iThenticate report* attached, especially the content with *red color marked 1*, and you can also add some appropriate citations to make sure that there is no large part repetition with the published paper. |
The manuscript has been modified to minimise the repetitions as suggested. Modified sections are highlighted along with the revisions in the manuscript. We hope this is acceptable but can make further changes if required. |
|
Round 2
Reviewer 2 Report
Comments and Suggestions for Authors
The paper is very much improved and I have no problem in recommending it for publication in Foods.